# Video In-context Learning: Autoregressive Transformers are Zero-Shot Video Imitators

**Wentao Zhang**[1,2,†,*] **, Junliang Guo**[3,†] **, Tianyu He**[3] **, Li Zhao**[3] **, Linli Xu**[1,2,‡] **, Jiang Bian**[3]

[1] School of Computer Science and Technology, University of Science and Technology of China
[2] State Key Laboratory of Cognitive Intelligence
[3] Microsoft Research Asia
`wentaoz@mail.ustc.edu.cn, linlixu@ustc.edu.cn,`
`{junliangguo,tianyuhe,lizo,jiang.bian}@microsoft.com`
`https://aka.ms/vid-icl`

## Abstract

People interact with the real-world largely dependent on visual signal, which are ubiquitous and illustrate detailed demonstrations. In this paper, we explore utilizing visual signals as a new interface for models to interact with the environment. Specifically, we choose videos as a representative visual signal. And by training autoregressive Transformers on video datasets in a self-supervised objective, we find that the model emerges a zero-shot capability to infer the semantics from a demonstration video, and imitate the semantics to an unseen scenario. This allows the models to perform unseen tasks by watching the demonstration video in an in-context manner, without further fine-tuning. To validate the imitation capacity, we design various evaluation metrics including both objective and subjective measures. The results show that our models can generate high-quality video clips that accurately align with the semantic guidance provided by the demonstration videos, and we also show that the imitation capacity follows the scaling law. Code and models have been open-sourced.

## 1 Introduction

Humans acquire skills through imitation, a fundamental aspect of learning demonstrated even in early childhood. For instance, when children observe how to hold a spoon, they not only learn this specific action but can also generalize it to grasp different spoons in varied contexts. These demonstrations are primarily conveyed through visual signals, which are abundant in our environment and provide intricate guidance. Similarly, for the development of general AI agents, the ability to learn from visual demonstrations and generalize to new situations is crucial and desirable. This capability to understand and imitate visual signals is not only appealing but also essential for advancing AI towards more human-like learning and interaction.

Previous works (Bar et al., 2022; Wang et al., 2023a;b; Bai et al., 2023) have studied *image in-context learning* in image-based visual tasks. By formulating image perception tasks such as segmentation and detection to supervised image pair demonstrations in delicately designed structures, e.g., a joint grid image (Bar et al., 2022) for MAE models (He et al., 2022) or a sequence (Bai et al., 2023) for Transformer models (Vaswani et al., 2017), they encourage the model to mimic the task and generate the prediction of a given image query. However, these works require explicitly formulating the training samples in source-target pairs, which restricts in-context inference to a supervised capacity. Furthermore, image-level generation struggles to produce coherent and contextually appropriate sequences.

In this paper, we introduce the **Vid**eo **I**mi**T**ator (**VidIT**) model, an autoregressive Transformer (Vaswani et al., 2017; Touvron et al., 2023) trained with the next token prediction objective

---

*Work done during an internship at Microsoft Research Asia in March 2024.

†Equal Contribution.

‡Corresponding author.

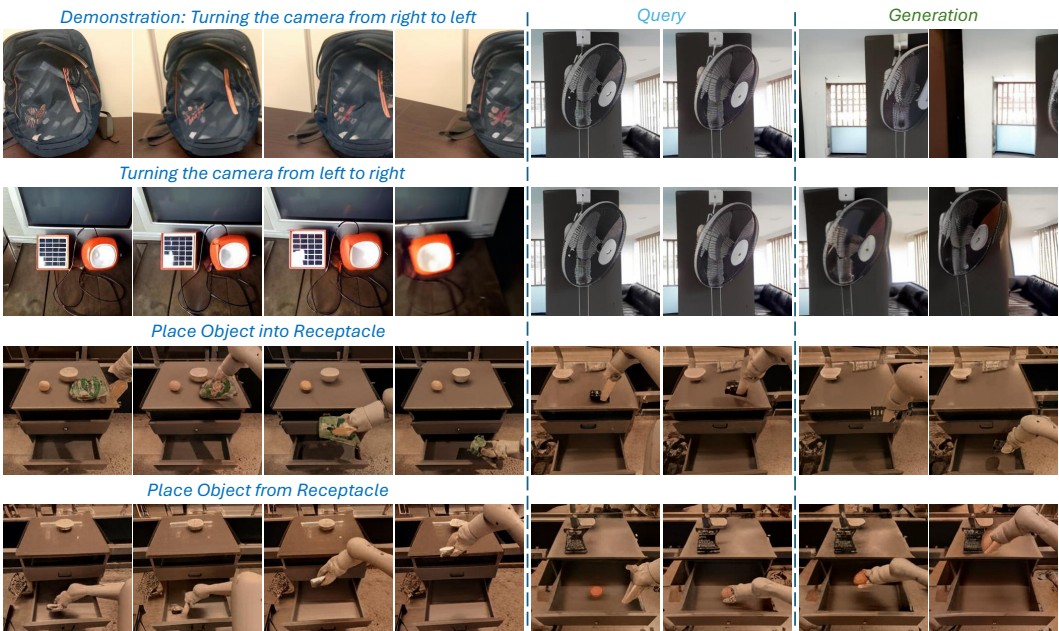

Figure 1: Illustrations of VidIT. Given a query video clip, the model generates different results based on the provided demonstration video clips. The generated results are coherent with the query in content and semantically consistent with the demonstrations. Text descriptions are annotated for clarity and not used as model input.

on video datasets. VidIT uses videos as the primary unit for both inputs and outputs, with each training sample consisting of a sequence of frames from a single video, encoded as discrete tokens using a vector-quantized encoder (Van Den Oord et al., 2017; Patil et al., 2024). The model learns to predict future frames based on preceding ones through self-supervised learning. Though demonstration-generation pairs are not provided during training, the autoregressive paradigm enables the model to discern the underlying structures and patterns within the videos, leading to the emergence of a zero-shot imitation capability. Consequently, when the model encounters demonstration videos during inference, it can generalize and apply these learned patterns to generate coherent and contextually appropriate video sequences. This zero-shot capability echoes findings observed in large language models (Brown et al., 2020).

Specifically, demonstration videos are highly versatile and capable of conveying a wide range of information, such as examples for tasks including object manipulation, or movements of the camera in an ego-centric video. Guided by these demonstrations, the model receives a query video clip set in a new scene and generates a subsequent video sequence that mimics the semantic actions from the demonstrations (see Figure 1 for examples). This allows VidIT to address multiple downstream tasks, such as embodied planning and simulating, by letting a query robot imitate the actions demonstrated by humans or other agents. Given that videos excel in describing low-level details (where language may fall short) and temporal dynamics (where images are insufficient), VidIT acts as a crucial interface for models to interact with the real world.

To comprehensively and accurately evaluate the model performance in VidIT, we develop both objective and subjective metrics to assess the generated videos in terms of visual quality, semantic accuracy, and consistency with the prompted demonstrations. Our extensive experiments demonstrate that the model not only produces high-quality video clips but also successfully adheres to the semantic guidance provided by the demonstration examples. In addition, we show that the zero-shot imitation capacity also follows the scaling law (Kaplan et al., 2020) of large models, illustrating the potential of future works. The main contributions of this work are summarized as follows:

- We propose and study the task of video imitation, which enables the model to interact with real-world demonstrations through video generation.

- We train a large Transformer model VidIT that exhibits powerful video imitation learning capacity, which also follows the scaling law of large models.

- We propose various evaluation metrics to evaluate the visual quality and semantic accuracy of generated videos, providing a solid benchmark for the evaluation of video imitation learning.

## 2 RELATED WORK

**Imitation Learning and In-context Learning**   Imitation Learning is a crucial paradigm in reinforcement learning, wherein the expected behavior is acquired by mimicking demonstration's actions (Zare et al., 2024). By learning on state-action pairs from an expert, the model is expected to map the current state to the corresponding actions. Such learning paradigm is close to human behavior and is appealing in interactive circumstances.

This ability to imitate has also been identified as a key feature in large language models (Radford et al., 2019; Brown et al., 2020; Touvron et al., 2023; Chowdhery et al., 2023), which is referred to as *in-context learning* in LLM area. By conditioning on a sequence of text templates, an LLM generates coherent content based on these templates. This paradigm obviates the need for parameter updates and thereby benefiting the downstream usage of LLMs (Brown et al., 2020; Touvron et al., 2023).

Different prompting designs (Wei et al., 2022; Guo et al., 2023; Li et al., 2023) empower LLMs to deal with a wide range of natural language understanding and generation tasks.

As for the in-context paradigm in vision area, current research mainly focus on teaching the model to mimic the vision task provided by demonstrations. Pioneering works construct vision imitation as an image inpainting (Bar et al., 2022; Wang et al., 2023a;b) task. Given multiple query-answer image pairs arranged in a grid image, models are optimized to reconstruct the masked answers under the MAE projective (He et al., 2022). Recently, LVM (Bai et al., 2023) flatten image pairs into a sequence and train an auto-regressive Transformer with next token prediction. These models show powerful capacity to imitating vision tasks such as semantics segmentation and detection from the demonstrations, but rely on training on supervised datasets. In addition, when extending the basic elements (i.e., demonstrations, queries and predictions) from images to videos, unfortunately, existing models struggle to handle the increased complexity as they are not designed to capture the spatial-temporal relationships of inputs. Specifically, though LVM (Bai et al., 2023) is able to generate consecutive frames of a query video clip, the video imitation capacity (e.g., generating different consequences with different demonstrations) is not demonstrated.

**Video Generation**   The studied video imitation learning can be considered as a conditional video generation problem, which has recently become a prominent focus in research. Text is the most commonly utilized condition, and various text-to-video models have shown promising results in generating high-fidelity videos (Ho et al., 2022b;a; Brooks et al., 2024; Hong et al., 2023; Villegas et al., 2023; Kondratyuk et al., 2023; Yu et al., 2023b). Recently, video generation models are spread to other related domains such as embodied AI (Yang et al., 2024), where they are utilized in visual planning (Du et al., 2023) and simulation (Yang et al., 2023) with actions and observations as conditions, illustrating the potential of this area. As for model architectures, Diffusion models (Ho et al., 2020; Nichol & Dhariwal, 2021) and Transformers (Vaswani et al., 2017) are both widely adopted in different scenarios, and we choose the autoregressive Transformer as the backbone model in this paper, as its abilities to model long-range dependencies and contextual relationships are crucial for video imitators.

## 3 VIDEO IMITATOR

We introduce the proposed Video Imitator in this section. We begin with the problem definition of both image and video imitation in Section 3.1, highlighting the differences between them. We then elaborate on the training and inference pipeline of VidIT in Section 3.2. Dataset selection is discussed in Section 3.3.

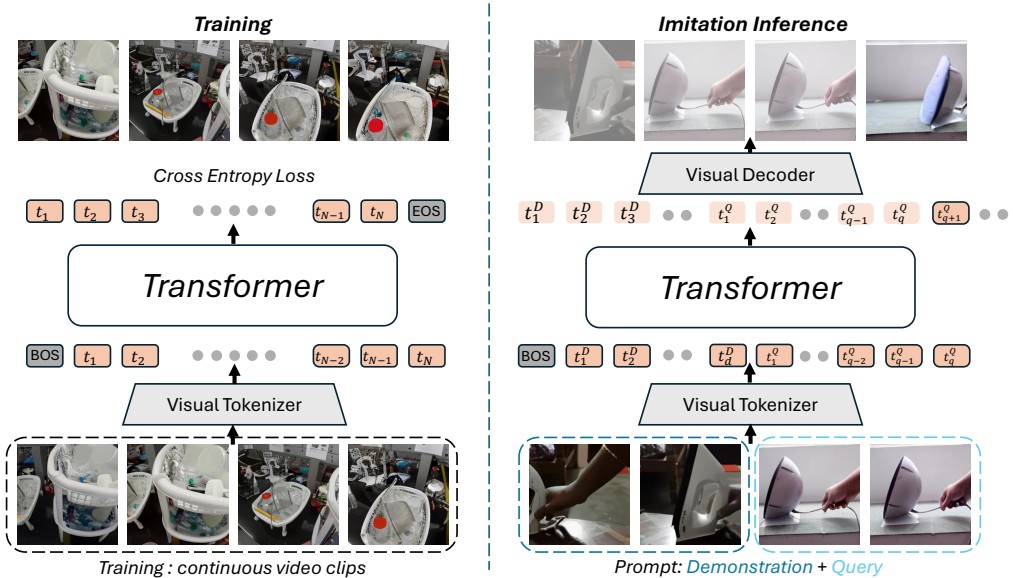

Figure 2: The framework of VidIT. *Left*: Training the VidIT. The data used for training are continuous video clips and the Transformer is trained by next token prediction objective. *Right*: Video Imitation Inference. The model is conditioned on demonstration videos and generates the subsequent frames of a given query video.

## 3.1 PROBLEM DEFINITION

Typically, the image task imitator models follow the setting akin to *question answering*: the demonstration is composed of $k$ image pairs $D^I = \{s_i^q, s_i^a\}_{i=1}^k$, where $s_i^q \in \mathbb{R}^{(3, n_r, n_r)}$ denotes the input image with resolution $n_r$ and $s_i^a$ denotes its target (usually an image with task-specific annotations (Bai et al., 2023)). Given a query image $x^I$, the model predicts its target $y^I$ conditioned on the demonstration:

$$f_\theta(x^I) = P(y^I|x^I, D^I), \tag{1}$$

where $f_\theta(\cdot)$ represents the utilized vision model such as an MAE-pretrained ViT (Wang et al., 2023a; Bar et al., 2022) or an auto-regressive Transformer (Bai et al., 2023), which is expected to discern the inherent task from image pairs $(s_i^q, s_i^a)$ and process $x$ accordingly.

Unlike image task imitators, VidIT takes videos as basic element, focusing on inheriting semantics from demonstrations rather than discerning specific tasks. Specifically, the demonstration $D^V = \{(s_i^1, \cdots, s_i^{n_i})\}_{i=1}^k$ comprises $k$ video clips, and each clip consists of $n_i$ frames. Given a query video clip $x^V = (s_q^1, \cdots, s_q^{n_q})$, the objective of VidIT can be formulated as:

$$f_\theta(x^V) = P(y^V|x^V, D^V), \tag{2}$$

where $y^V = (s_y^1, \cdots, s_y^{n_y})$ denotes the generated video clip, which should be perceptually coherent with the query $x^V$ while semantically consistent with the demonstration $D^V$ at the same time.

## 3.2 APPROACH

Training a Transformer for vision tasks typically consists of two stages, 1) training a visual tokenizer such as VQ-VAE (Van Den Oord et al., 2017; Esser et al., 2021) to convert each image to discrete tokens; 2) each training sample is constructed as a sequence of tokens to train the Transformer decoder. For the first stage, we utilize a public pretrained checkpoint (Patil et al., 2024) with $16\times$ spatial compression rate as the VQ tokenizer. Our framework also fits for recent tokenizers with both spatial and temporal compressions (Yu et al., 2023a;b), and we leave it for future work. Formally, for a training video clip $x^V = (s^1, \cdots, s^n)$ with $n$ frames, the VQ tokenizer converts it to a flat sequence $x^t = (t_1, \cdots, t_{n_t}, t_{n_t+1}, \cdots, t_N)$, where $n_t$ denotes the number of quantized ids that represents one frame, and $N = n \cdot n_t$ denotes the total length of the sequence. We append special

tokens `[bos]` and `[eos]` to the front and end of the sequence, and no special tokens are inserted between sequences of different frames.

Then, in the second stage, we follow the architecture of LLaMA (Touvron et al., 2023) utilizing RMSNorm normalizing (Zhang & Sennrich, 2019) and Rotary Embeddings (Su et al., 2024), and train a Transformer decoder in an autoregressive way:

$$f_\theta(x^t) = \prod_{i=1}^{N} P(t_i|t_{<i}), \tag{3}$$

where the model predicts each token conditioned on previous ones. Note that in the training stage, each sequence is sampled from one original video, and we do not concatenate video clips from different videos.

**Imitation Inference** The imitation inference format differs from that in the training. Following Equation (2), a number of video clips are selected as the demonstration $D^V$, which is appended in front of the query clip $x^t_{<j} = (t_1, \cdots, t_{j-1})$ to let the model generate corresponding responses:

$$f_\theta(x^t_{>j}) = \prod_{i=j}^{N} P(t_i|x^t_{<i}, D^V). \tag{4}$$

The generation results $x^t_{\geq j} = (t_j, \cdots, t_N)$ are vector-quantized IDs, which are then fed to the pretrained VQ decoder to reconstruct as images. We provide an illustration of training and inference pipelines in Figure 2.

**Zero-shot Capacity** The training and inference processes of the model differ in that the demonstration $D^V$ is not provided during training. Despite this, the model exhibits zero-shot video imitation capabilities. This can be attributed to two key reasons. Firstly, no special separation token is inserted between frames, allowing the preceding sequences $t_{<i}$ in Equation (3) to be implicitly viewed in a demonstration-query format, i.e., $P(t_i|t_{<i,>j}, t_{\leq j})$ where $j < i$. This implies that the model has inherently learned to handle sequences resembling the demonstration-query structure during training. Secondly, the autoregressive nature of the Transformer enables it to seamlessly extend its sequence prediction capabilities to scenarios where the demonstration and query come from different videos, thus facilitating smooth generalization to the imitation paradigm. Our experiments further show that training the model with explicit imitation examples does not yield significant improvements over the zero-shot approach. Refer to Sec 5.1 for more discussions.

**Incorporate Other Modalities** We mainly focus on videos as the demonstration in this paper, but our approach can be easily extended to acquiring other modalities such as text. To do so, we just need to transfer original text descriptions into latent representations denoted as $c$ by a pretrained language model (Raffel et al., 2020), then take $c$ as an additional condition while training the Transformer (Equation (3)) as well as imitation inference (Equation (4)). We show in experiments Sec 5.5 that our model understands and reacts to both text and video demonstrations.

### 3.3 DATA

While large language models are trained on vast amounts of data (usually trillions of tokens (Hoffmann et al., 2022; Touvron et al., 2023)), high-quality video data is constrained. In addition, VidIT prefers videos that exhibit not only rich content but also clear causal relationships and interactivity. As a result, among various public video datasets, we focus on these accomplish embodied tasks and select two primary datasets as our main training data sources: 1) **Ego4d** (Grauman et al., 2022), an egocentric video dataset featuring abundant first-person activities; and 2) **Kinetics-600** (Carreira et al., 2018), a comprehensive video dataset comprising diverse human activities.

Additionally, we incorporate self-collected videos that contains a large amount of general real-world videos, to augment the variety of video content.

To validate the VidIT's imitation capability, we choose the **Something-Something v2** (SSv2) as the main evaluation dataset (Goyal et al., 2017) where each video depicts a basic action that occurs in the

physical world, such as moving objects in a direction. These videos convey strong semantic information, which can be utilized to construct the demonstrations. We utilize the evaluation split of SSv2 as the evaluation set for all experiments. In addition, we include the **Robotics Transformer-1** (RT-1) (Brohan et al., 2022) dataset and **MineRL**[1] to demonstrates VidIT's effectiveness on embodied AI and interactive tasks.

## 4 EXPERIMENTAL SETUPS

We design tailored evaluation pipelines to assess the VidIT's capacity to perform video imitation. We introduce the proposed evaluation metrics in Section 4.1, and summarize the details of our implementations in Section 4.2.

### 4.1 EVALUATION

The evaluation of VidIT should include two aspects, the first is the visual quality as a normal video generation task, and the second is semantic accuracy which is more critical as it reflects whether the model understands and follows the semantic guidance provided by demonstrations. To validate semantic accuracy, we design four kinds of demonstrations.

- **No Demonstration.** This setting is akin to unconditional video prediction, where the model is asked to complete the query $x^V$ without any demonstration.
- **Random Demonstration.** In this type of demonstration, video clips are chosen from arbitrary videos within the entire SSv2 dataset.
- **In-class Demonstration.** Given a video clip query, the demonstrations are sampled from videos with the same action label as the query.
- **Contrastive Demonstration.** In contrast to the previous one, the sampled demonstrations have the contrast action label to the query.

We then evaluate the model in two settings, 1) using ground truth query labels to evaluate results, to test whether the demonstrations enhance or deviate from the generation results; 2) using demonstration labels, to assess whether the model accurately acquires the semantic guidance and generates corresponding results. We define various metrics to deal with these two settings.

**Automatic Metrics**   For visual quality, we adopt several widely utilized metrics including **PSNR** and **LPIPS** (Zhang et al., 2018) to perform a frame-by-frame comparison between the generation results and the ground truth video clip, i.e., the consequences of the query. Additionally, we include the **FID** (Heusel et al., 2017) and **FVD** (Unterthiner et al., 2018) score to quantify the distribution difference. These metrics are suitable for the first setting where ground truth is available.

For semantic accuracy, we propose two classification-based metrics:

- **Video Accuracy** (V-Acc). We utilize an off-the-shelf video classifier (Tong et al., 2022) trained on the SSv2 dataset to calculate the classification accuracy of the generated video clips. Specifically, we predict the action label of each generated video clip and compare it with the ground truth label of the query. This metric provides a perceptual assessment of the semantic information in generation results. In particular, we use the `videomae-base-finetuned-kinetics`[2] checkpoint.
- **Probing Accuracy** (P-Acc). While V-Acc relies on a pretrained classifier and judges on visual signals, we propose another metric named P-Acc to operate on the latent representation of VidIT. To directly validate the semantic information contained in latent representations, inspired by the widespread use of probing in vision feature extractors (He et al., 2022; Huang et al., 2023; Bardes et al., 2023), we train a probing classifier which takes hiddens in the last Transformer layer of the generation results as input and predicts the corresponding action label.

---

[1] https://github.com/minerllabs/minerl
[2] https://huggingface.co/MCG-NJU/videomae-base-finetuned-kinetics

Most automatic metrics are only suitable for the first setting where ground truth video is available. For the second setting without ground truth, only the probing accuracy can be utilized. As a supplementary, we also introduce human evaluation.

**Human Evaluation**  We manually select a subset from the SSv2 validation set, by picking out action labels with contrastive semantics, such as *Pulling [something] from left to right* and *Pulling [something] from right to left*. For a query video clip, we sample two demonstrations from the same and contrastive class respectively, constructing a pair of evaluation samples (just as shown in Figure 1). We involve 10 experienced users to score 20 pairs of samples from several aspects, i.e., visual quality, semantic alignment and control. Alignment indicates whether each generation is semantically consistent with the demonstration, while control justifies whether the pair of results perform contrastive behavior following the demonstrations. Participants were presented with a pair of videos at a time and asked to rate each video for each score on a scale of 1 to 5. We calculated the average score as the final result.

## 4.2 IMPLEMENTATION DETAILS

**Preprocessing**  For each dataset, we assign a stride to sample frames at regular intervals in order to capture human-recognizable video clips. The stride varies among datasets depending on the average FPS but remains consistent within each dataset. Subsequently, we resize and center-crop the images to a fixed size of $256 \times 256$ resolution.The pretrained VQ-GAN tokenizer (Patil et al., 2024) takes in $256 \times 256$ sized images and produces $16 \times 16 = 256$ discrete tokens with a compression coefficient $f = 16$. Tokens of different frames are concatenated in the order of original frames, and each training sequence contains 16 images and 4096 discrete tokens.

**Transformer Architecture**  we adopt the LLaMA (Touvron et al., 2023) architecture, a typical decoder-only transformer model for auto-regressive modeling with designs advantageous to large-scale modeling. we experiment with different sets of hyper-parameters which result in models with 300M, 700M and 1.1B parameters separately. The details of hyper-parameter settings are shown in Table 8. We introduce more training details in Appendix B.3

**Inference**  In our main experiments, we set $k = 1$ to restrict the demonstration to one video. In this way, the demonstration clip contains $8$ frames, while the query and generation results both consist of $4$ frames. We also study different compositions such as $k = 2$ demonstrations sampled from different videos and each with 4 frames. Please refer to Section 5.5 for more discussions.

**Baselines**  We compare the proposed VidIT model with recent visual generative models based on auto-regressive Transformer. **LVM** (Bai et al., 2023) is a natural baseline that shows strong image imitation capacity. **DeLVM** (Guo et al., 2024) is a data efficient version of LVM, which involves extra knowledge distilling step and have fewer parameters. **LWM** (Liu et al., 2024) is designed to understand and generate long-context video contents, which is trained on aligned text-video pairs. To evaluate the generalizability of our model, we train several variants: *Pretrain*, which is trained without SSv2 in the training set; *Pretrain w/ In-domain finetune*, which is fine-tuned on SSv2 from the *Pretrain* model where each sample consists of continuous 16 frames, similar to the pretraining stage; and *Pretrain w/ Imitation finetune*, which is fine-tuned using the same data format as imitation inference, consisting of two 8-frame videos from the same class.

## 5 RESULTS AND ANALYSIS

## 5.1 MAIN RESULTS

If not specified, we use our largest 1.1B model to generate results. We first evaluate the VidIT results using the ground truth query label. In this setting, all automatic metrics can be utilized. The quantitative results are presented in Table 1, where each row corresponds to a different demonstration type. We draw the following conclusions from the results.

**In-class Demonstrations Contribute to Stronger Semantic Accuracy**  Our analysis begins by focusing on the result of the *Pretrain* model in the first block. We observe a significant enhancement

Table 1: The Semantic Accuracy metrics and Visual Quality metric of different demonstration type and training strategies. The best result in each block is **bolded** and second best result is underlined.

| Demonstration | Semantic Accuracy | | Visual Quality | | | |
|---|---|---|---|---|---|---|
| | V-Acc ↑ | P-Acc ↑ | PSNR ↑ | LPIPS ↓ | FID ↓ | FVD ↓ |
| *Pretrain* | | | | | | |
| No Demonstration | 22.9 | 29.6 | **13.20** | **0.442** | **20.94** | **119.51** |
| Random Demonstration | 22.7 | 28.3 | 13.01 | 0.450 | 22.53 | 131.34 |
| In-class Demonstration | **24.7** | **36.7** | 13.07 | 0.447 | 21.95 | 125.77 |
| *Pretrain w/ Imitation Finetune* | | | | | | |
| No Demonstration | 24.2 | 26.7 | 13.02 | 0.460 | 22.21 | 129.68 |
| Random Demonstration | 23.1 | 25.6 | 13.01 | 0.454 | 20.66 | 115.76 |
| In-class Demonstration | **25.7** | **40.7** | **13.08** | **0.450** | **20.49** | **113.52** |
| *Pretrain w/ In-domain Finetune* | | | | | | |
| No Demonstration | 25.0 | 30.8 | 13.09 | 0.446 | 20.87 | 108.12 |
| Random Demonstration | 24.9 | 34.1 | 13.16 | 0.440 | 19.17 | 95.72 |
| In-class Demonstration | **25.9** | **48.8** | **13.21** | **0.438** | **18.92** | **88.63** |

Table 2: Results on automatic and human evaluation of VidIT and baseline models.

| Baselines | Automatic Metrics | | Human Evaluation | | |
|---|---|---|---|---|---|
| | P-Acc | PSNR | Quality | Alignment | Control |
| LVM (Bai et al., 2023) | 27.1 | 12.45 | 3.50 | 2.71 | 1.81 |
| DeLVM (Guo et al., 2024) | 32.8 | 12.69 | 3.21 | 2.28 | 2.11 |
| LWM (Liu et al., 2024) | 24.9 | 11.89 | 3.39 | 2.33 | 2.05 |
| **VidIT** (ours) | **38.5** | **13.07** | **4.12** | **3.73** | **3.01** |

in the semantic accuracy when using in-class video clips as demonstrations, compared to without demonstration or using randomly selected ones. Specifically, there is a notable 7.1% / 8.4% improvement in P-Acc and a 1.8% / 2.0% gain in V-Acc over no or prompting with random demonstrations respectively. These findings validate our proposition that when prompted with semantically related demonstrations, the model more accurately generates video clips that adhere to the original trace, in a zero-shot way on both the domain and the imitation ability.

**Random Demonstrations Mislead the Semantics**    Beyond the inferior performance compared to in-class demonstrations, random ones also reveal a gap of 0.2% in V-Acc and 0.7% in P-Acc to the no demonstration setting. This indicates that prompting with unrelated demonstrations negatively impacts the semantics of the generated results.

**In-domain Finetuning Leads to General Improvement**    We further analyze the results presented in the second and third blocks of Table 1, focusing on different fine-tuning strategies. The superiority of in-domain finetuning over the initial *Pretrain* is evident, indicating a straightforward performance boost when in-domain training data is available. In addition, when comparing in-domain finetuning with imitation finetuning, the in-class P-Acc of the former rises to 48.8%, showcasing an 8.1% improvement over the latter. These findings underscore the remarkable *zero-shot capacity* of VidIT, showcasing the model's capability at adapting to the imitation formatting through exclusive training on continuous video clips, while explicit finetuning with the target format fails to yield significant enhancements.

**Evaluation with Demonstration Label**    We evaluate the generation results with the demonstration label to assess the controllability of the model. We use results with both in-class and contrastive demonstrations as described in Section 4.1 to train the probing model. As shown in Table 3, both cases have well above average probing accuracy, indicating that the representations of generation results successfully obtain the semantic information from demonstrations.

Table 3: The Probing accuracy evaluated by demonstration labels.

| Demonstration | P-Acc |
|---|---|
| Contrastive | 35.5 |
| In-class | 33.8 |
| Random | 16.7 |

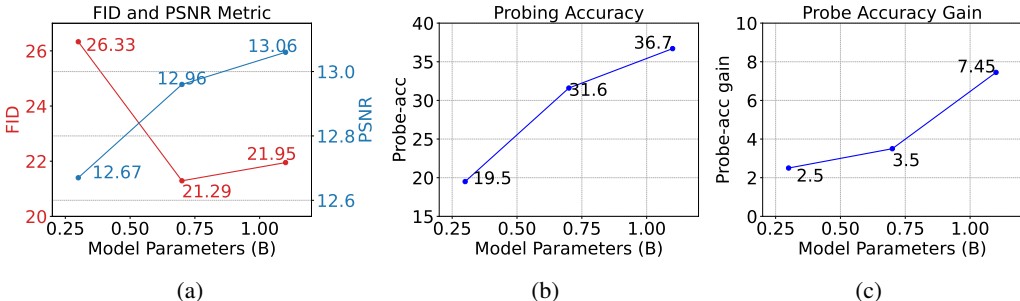

Figure 3: The performance when scaling model parameters.

## 5.2 COMPARISON WITH BASELINE MODELS

We compare our VidIT with baseline methods mentioned in Section 4.2 on the video imitation abilities.

We report both automatic metrics and human evaluation results in Table 2. The full evaluation of baseline models is reported in Table 6 in Appendix A.6. As shown in the results, our VidIT model significantly outperforms the baseline models in both objective and subjective evaluations. We attribute this difference to the training data formats of VidIT and other models. LVM and DeLVM mainly relies on pairs of images and their annotations, which limits their ability to extract semantic information across consecutive frames and generate coherent sequences correspondingly. Moreover, the training process of LWM are not designed to capture the demonstration's semantics, leading to degradation on imitation performance.

## 5.3 SCALING BEHAVIOR

As describe in Section 4.2, we set up different sizes of model, respectively 300M, 700M and 1.1B. In this section, we assess the models' scaling behavior from the following aspects.

**Scalability on Visual Quality** We present the PSNR and FID scores of models in various sizes in Figure 3a. The results indicate that larger models tend to produce samples of higher quality.

**Scalability on Semantic Accuracy** We present both the absolute probing accuracy scores of in-class demonstrations and the performance gain over random demonstrations in Figure 3b and 3c. From both results, we observe a clear trend that larger models yield more informative hidden representations, and offer more precise control over the generation results when conditioned on different demonstrations.

## 5.4 GENERALIZATION ABILITY

In this section, we elaborate on the generalization ability of the VidIT model. Thanks to its self-supervised training paradigm, VidIT can be smoothly adapted to multiple scenarios and tasks with minimal modifications.

Firstly, we apply VidIT to Robotics Transformer dataset (Brohan et al., 2022) to imitate actions of a robotic arm, such as opening and closing drawers and placing objects. The samples on Figure 4 demonstrates that VidIT can successfully mimic various robotic arm actions from demonstrations, completing the frame sequences with the same movements.

Additionally, we highlight that VidIT can serve as an underlying video generation engine within an interactive agent. In this setup, VidIT receives previous observations and generates subsequent frames, which are then converted to action signals via the inverse dynamics mechanism. This process ensures that the action signals reflect VidIT's imitation intent. As depicted in Figure 6 in Appendix A.1, agents perform specific actions guided by VidIT, showcasing the zero-shot imitation ability of VidIT and revealing its potential to interact with the environment.

We also validate VidIT's capability as a video task imitator. Following the approach of image task imitators like LVM, we create video segmentation exemplars to form QA pairs on the VIPSeg (Miao et al., 2022) dataset. The results, presented in Figure 7 in Appendix A.2, indicate that VidIT success-

fully learns the task format from the demonstrations. Moreover, we study how the model emerges the imitation ability by analyzing the attention score in Appendix A.5

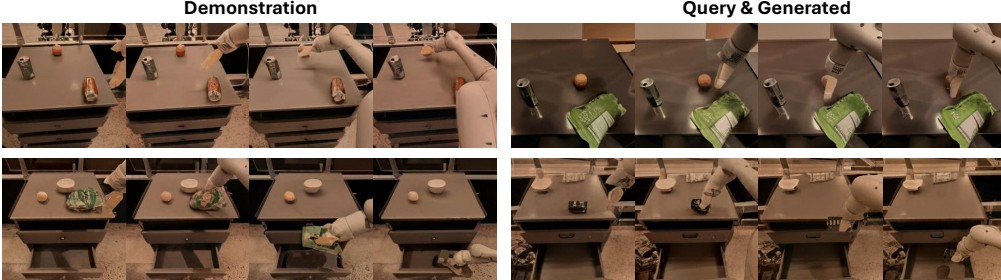

Figure 4: Generation Samples of VidIT on RT-1 Dataset.

## 5.5 ABLATION AND ANALYSIS

**Different Number of Demonstrations**  We investigate the impact of different numbers of demonstrations on VidIT performance. Within a 16-frame context window, we fix the query length as 4 frames and the generation as 4 frames, and construct four demonstration formats: 1) no demonstration; 2) one 4-frame demonstration; 3) two 4-frame demonstrations sampled from different videos; 4) one 8-frame demonstration. The results in Table 4 demonstrate that providing demonstrations generally improves both P-Acc and V-acc scores. Additionally, providing more ($2 \times 4$ versus $1 \times 4$) or longer demonstrations ($1 \times 8$ versus $1 \times 4$) both result in better performance.

Table 4: Semantic accuracy of different demonstration formats.

| Demonstration | P-Acc | V-Acc |
|---|---|---|
| No | 29.6 | 22.9 |
| $1 \times 4$ frames | 35.6 | 22.9 |
| $2 \times 4$ frames | **37.2** | 24.2 |
| $1 \times 8$ frames | 36.7 | **24.7** |

Table 5: Results with text as demonstrations.

| | V-Acc | FID |
|---|---|---|
| Tune w/ text | **28.8** | **19.46** |
| + Infer w/o text | 24.6 | 19.72 |
| Tune w/o text | 24.7 | 20.02 |

**Text as Demonstrations**  To demonstrate the versatility of our model, we explore incorporating text into the demonstrations. We fine-tune the 300M *Pretrain* model on the SSv2 dataset by appending corresponding text annotations to the front of each video. The text annotations are encoded into latent representations using a pretrained T5-large model (Raffel et al., 2020). The results, presented in Table 5, indicate that our model can seamlessly adopt demonstrations in various modalities. Additionally, adding textual descriptions enhances the model's imitation capacity.

## 6 CONCLUSION

In this paper, we introduce Video Imitator, a novel model that extends imitation learning to video data. By converting video frames into sequences of discrete tokens and training the autoregressive Transformer with next token prediction, our VidIT model exhibits zero-shot imitation capabilities, which can generate semantically coherent and contextually appropriate video sequences based on provided demonstrations. Extensive experiments confirm that VidIT effectively captures and conveys the semantic information embedded in the demonstrations, demonstrating its potential to enhance various downstream tasks such as visual planning. Additionally, the model's versatility is verified by its ability to understand and integrate multi-modal demonstrations simultaneously.

## ACKNOWLEDGMENT

This research was supported by the National Natural Science Foundation of China (Grant No. 62276245).

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

# A ADDITIONAL RESULTS

In this section we provide more samples on different video datasets and benchmarks that VidIT model produces. Synthesis videos on Something-something v2 dataset are presented in Table 9. One can also navigate to our project page https://aka.ms/vid-icl for clearer presented samples. Table 10 includes samples on Robotics Transformer-1 dataset, showcasing the ability of VidIT to fit on various form of video datasets. Furthermore, VidIT as a versatile generalist, can act as a simulator in reinforcement learning tasks, for which we provide detailed explanations.

## A.1 VIDIT AS SIMULATOR

We demonstrate that VidIT can also function as a simulator in reinforcement learning tasks by evaluating it on the $VP^2$ (Tian et al., 2023) benchmark. $VP^2$ is a benchmark for video prediction models used in robotic manipulation via model-predictive control. VidIT generates future frames with videos that accomplishing the same task as the demonstration, where the generated frames inversely reflect the corresponding actions that correctly interact with the environment (i.e. inverse dynamics). We compare the trajectories produced by the SVG (Villegas et al., 2019) baseline and VidIT in Figure 5. Evaluating on the **Push-red** task, we find that VidIT provides more precise control over the environment interaction.

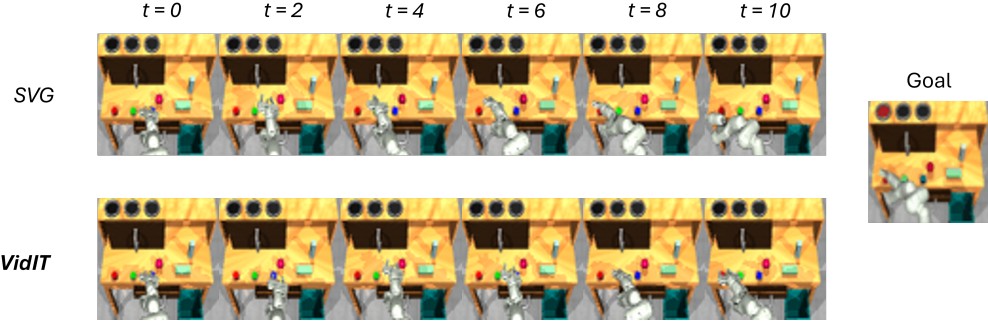

Figure 5: The trajectory produced by SVG and VidIT respectively.

Additionally, we evaluate VidIT on the MineRL environment as the underlying video generation model, where VidIT take in previous game scenes as conditions to imitate and generate subsequent scenes. Generated game scenes are then fed into inverse dynamics models to generate the next action signal. We showcase two types of actions: **Chop wood** and **Attack sheep**. In Figure 6, the VidIT correctly generate game scenes and produce corresponding action signal to finish the action.

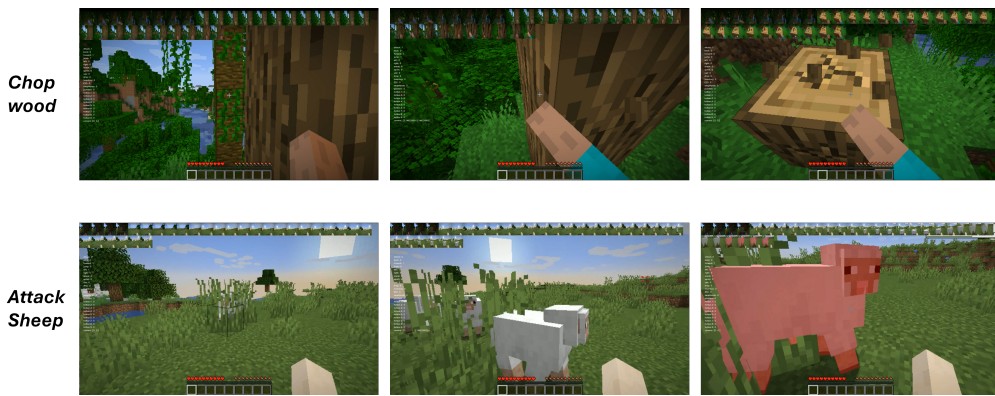

Figure 6: The VidIT model functions as video generator for *chop-wood* and *attack-sheep* in MineRL.

## A.2 VIDIT AS TASK IMITATOR

We expand the role of VidIT model to the task imitator on video, which mirrors the cases in image imitation learning. In this section, we finetune VidIT in VIPSeg dataset (Miao et al., 2022) for a few steps and show that VidIT can also smoothly generalize to video perceptual tasks like video segmentation. As depicted in Figure 7, VidIT successfully learns the task format from demonstration, and generates the segmentation outputs of the query video.

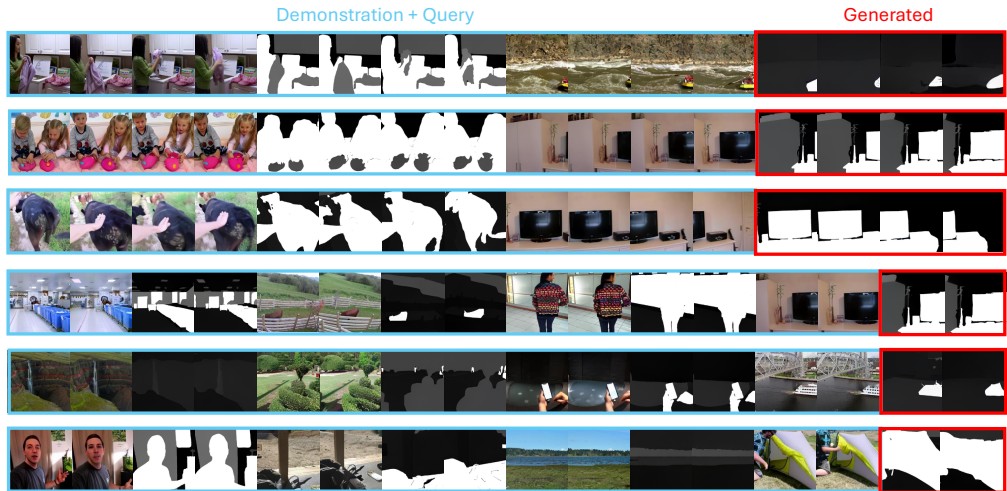

Figure 7: The VidIT can also imitate video perceptual tasks such as video segmentation.

## A.3 CASE STUDY: BASELINE METHODS

In this section we compare the generated cases of VidIT with the baseline methods. As in Figure 8, we see that conditioned on the same frames, VidIT is capable of generating more semantically accurate and high-quality video than baseline models. Moreover, VidIT is lower in parameter budget, highlighting the priority of VidIT as a video imitator model.

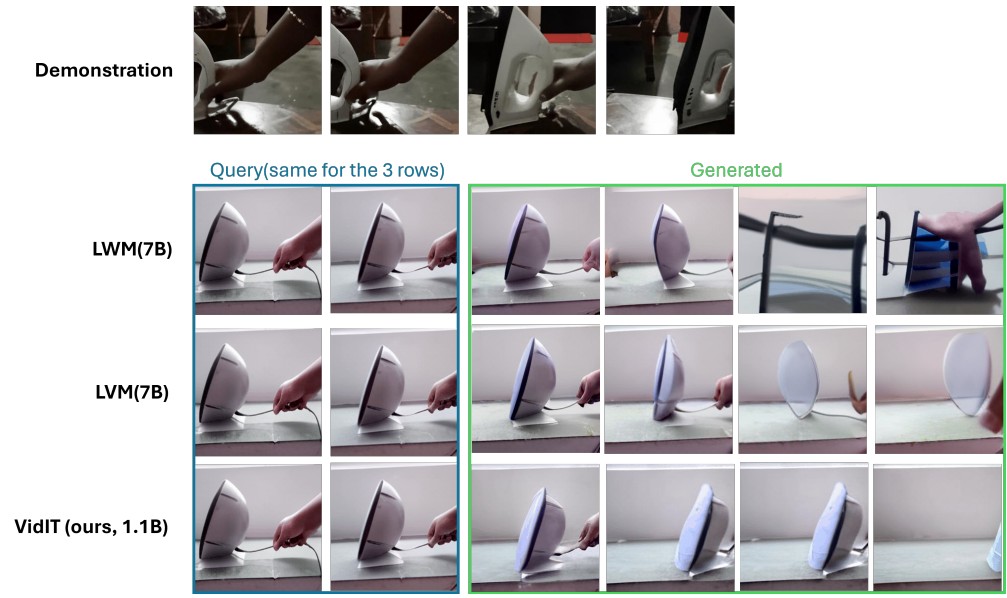

Figure 8: Generated cases of VidIT and baseline methods.

### A.4 FAILURE CASES

When prompted with semantically unrelated videos, VidIT may struggle to generate coherent content. In this section, we present failure cases where VidIT fails to align the query video with the semantic context of the demonstration video (see Figure 9).

For instance, in the first row, the demonstration video conveys the semantic information of moving a ball downward. However, when starting with a query video depicting an unrelated action, such as folding a blanket, the model struggles to reconcile the two contexts. As a result, it tends to ignore the demonstrated actions and generates content that lacks coherence with the demonstration.

Similar patterns are observed in other cases. These examples reveals VidIT's potential difficulty in maintaining semantic coherence when prompted with unrelated or ambiguous demonstrations. Addressing this limitation presents a promising direction for future research.

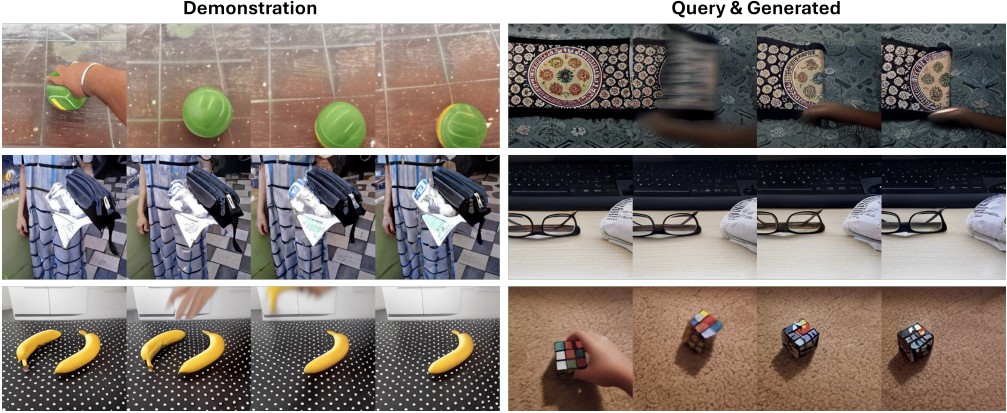

Figure 9: Failure cases of VidIT.

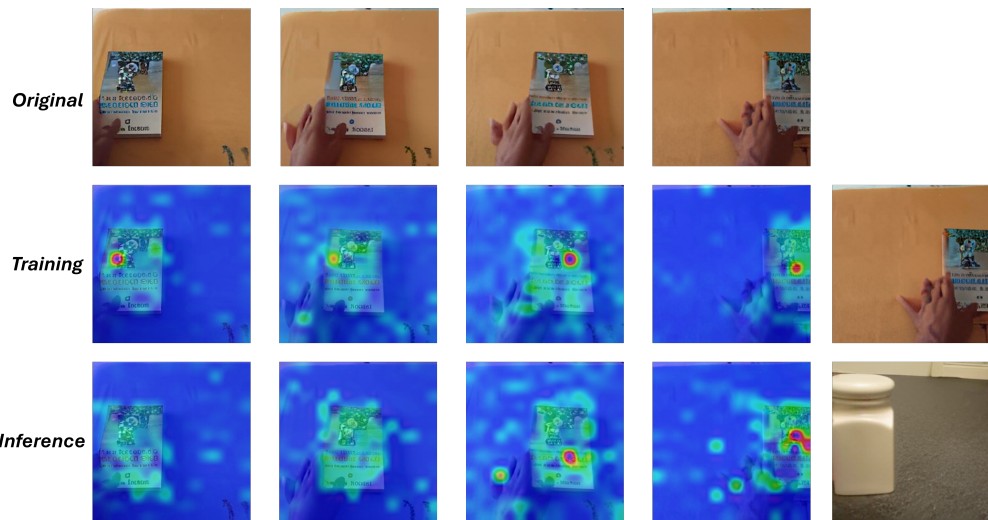

Figure 10: The attention weight visualization during training and inference respectively.

### A.5 INTERPRETABILITY OF GENERALIZATION

In this section, we highlight the interpretability of VidIT and its ability to generalize to imitation tasks in a zero-shot manner. To demonstrate this, we analyze the attention weights from the language model backbone to compare the similarity of attention patterns when provided with the same demonstration during training and inference.

Specifically, we extract the attention weights from the last layer of the model and average them across all attention heads. Since the tokens are flattened from the original frames in raster scan order, we rebind the tokens to their corresponding regions in the frames for better visualization. The results are shown in Figure 10.

The first row depicts the original frame sequence. The second row represents the training case, where frames are drawn from a single video clip. The third row illustrates the inference case, where the sequence begins from a different query frame. When generating a new token, we observe similar attention patterns across the demonstration frames. Notably, the attention is concentrated around the moving object within the frames.

This phenomenon indicates that the model effectively learns and leverages the semantic information from previous frames, enabling it to perform imitation tasks even when starting from a different scene.

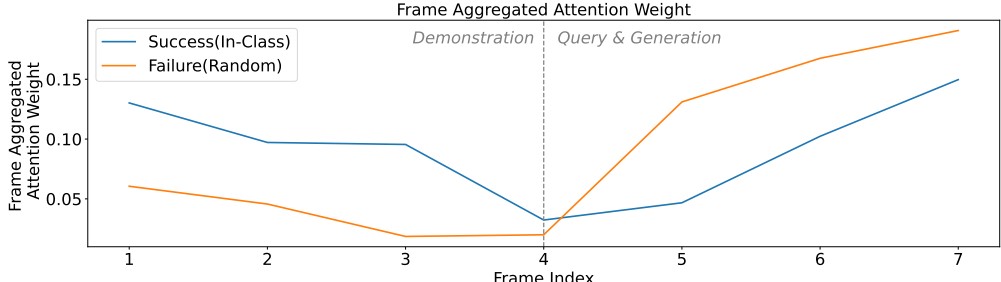

Figure 11: The aggragated attention score distribution in demonstration and query frames.

Additionally, we analyze the failure cases in detail from the perspective of attention weights. Specifically, we examine how new frames attend to previous frames during generation. Intuitively, when a new frame is effectively guided by the demonstration, it tends to allocate more attention to the frames in the demonstration video, rather than only attending to the query frames.

To illustrate this, we visualize the attention distribution of generated frames on previous frames, as shown in Figure 11. Clearly, successful generation samples exhibit a tendency to allocate more attention to the demonstration frames, indicating an ability to comprehend and utilize its semantic information. In contrast, the attention curve for failure cases reveals a higher focus on the query frames, effectively ignoring the guidance provided by the demonstration part.

## A.6 FULL EVALUATION METRICS ON BASELINES

In this section, we complete Table 2's results with the all the evaluation metrics in Section 4.1. From the results in Table 6, it is evident that the VidIT model consistently outperforms the baseline models in video imitation tasks, reinforcing its superior capability in capturing both semantic alignment and visual quality.

Table 6: Full automatic metrics for baseline models. The best result is **bolded** .

| Demonstration | Semantic Accuracy | | Visual Quality | | | |
|---|---|---|---|---|---|---|
| | V-Acc ↑ | P-Acc ↑ | PSNR ↑ | LPIPS ↓ | FID ↓ | FVD ↓ |
| LVM (Bai et al., 2023) | 21.2 | 27.1 | 12.45 | 0.489 | 27.58 | 163.13 |
| DeLVM (Guo et al., 2024) | 24.1 | 32.8 | 12.69 | 0.462 | 23.65 | 132.09 |
| LWM (Liu et al., 2024) | 21.8 | 24.9 | 11.89 | 0.474 | 29.91 | 191.56 |
| **VidIT** (ours) | **25.9** | **38.5** | **13.07** | **0.450** | **22.30** | **126.32** |

Table 7: The configuration of different size of models.

|      | Hidden dim | MLP dim | Num. Heads | Num. Layers |
|------|-----------|---------|------------|-------------|
| 300M | 1024      | 2688    | 8          | 22          |
| 700M | 1536      | 4096    | 16         | 24          |
| 1.1B | 2048      | 4096    | 16         | 26          |

Table 8: Model hyperparameters.

| Hyperparameter | Value |
|----------------|-------|
| Learning rate scheduler | inverse sqrt |
| Learning rate | $5e^{-4}$ |
| Warm up steps | 10000 |
| Weight decay | 0.01 |
| Optimizer | AdamW |
| AdamW betas | $(0.9, 0.95)$ |
| Context length | 4096 |

## B  IMPLEMENTATION DETAILS

### B.1  MODEL ARCHITECTURE

We utilize LLaMA as the Transformer architecture in our work. To validate the scaling behavior, we train three different sizes of the VidIT model within the LLaMA architecture: 300M, 700M, and 1.1B. We carefully tune the hidden dimension, MLP intermediate dimension, and the number of Transformer decoder layers to achieve different model sizes. The configuration details of these models are presented in Table 7.

### B.2  HYPERPARAMETERS

The hyperparameters used to train the VidIT model are presented in Table 8. We utilize inverse square root scheduler and start model training with 10,000 warmup steps.

### B.3  TRAINING DETAILS

Our largest variant, VidIT 1.1B, is trained on 2x8H100 nodes, with Pytorch DDP parallel strategy integrated in the Pytorch-Lightening trainer. The training time is about 3 hours per epoch and thus the total training GPU hours is 720 (15 epochs).

## C  LIMITATIONS

Despite the promising results, several limitations exist in this study. Firstly, while VidIT excels at generating short video sequences, its performance on longer sequences has not been thoroughly evaluated. Secondly, the model's dependence on the quality and relevance of provided demonstrations can lead to inconsistencies, particularly when demonstrations are noisy or semantically misaligned. Future work should address these limitations by utilizing visual tokenizers with temporal compression, and scaling or developing robust models to handle imperfect demonstrations.

## D  ETHICS STATEMENT

Positive societal impacts of VidIT include its potential to enhance various downstream applications such as embodied planning and robotic simulation by enabling models to imitate actions demonstrated in videos. As for negative societal impacts, the widespread adoption of this technique may raise concerns about privacy when powerful models gain the ability to analyze and generate video

content at scale. The datasets utilized in this study are publicly available and have been thoroughly reviewed to ensure they do not include personally identifiable information or offensive content. Nonetheless, as these datasets are sourced from the Internet, there may still be inherent biases. To address this, we have implemented a rigorous filtering process on the training data to minimize the potential for the model to generate inappropriate content.

Table 9: More samples of VidIT generated samples on Something-something v2 dataset.

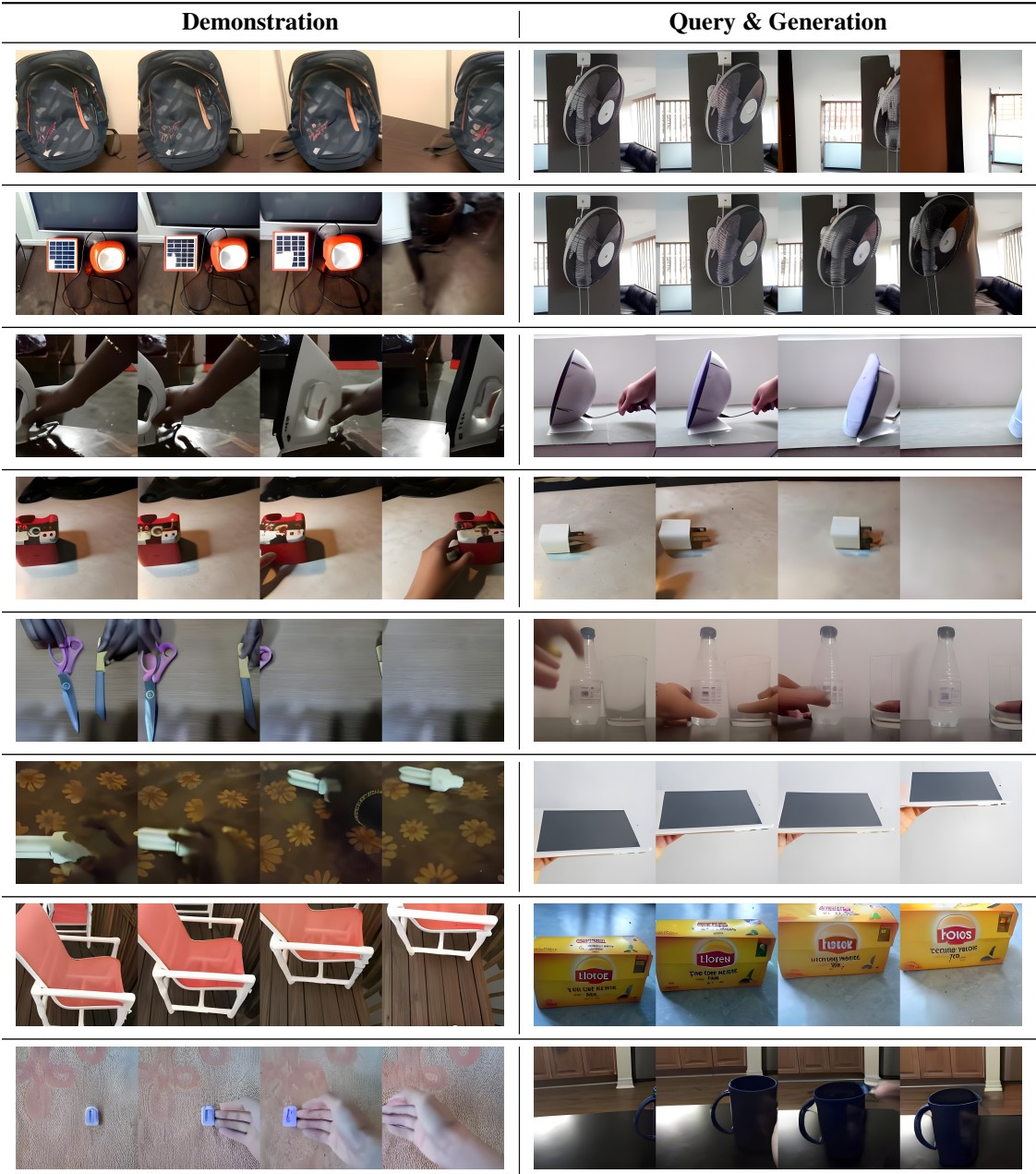

Table 10: More examples of VidIT generated samples on RT-1 dataset.

| Demonstration | Query & Generation |
| --- | --- |
|  |  |

