# OpenReview forum: "Video In-context Learning: Autoregressive Transformers are Zero-Shot Video Imitators"
_ICLR.cc/2025/Conference — ICLR 2025 Poster_

### Official Review · Reviewer_PhVw · 2024-10-16

**Soundness:** 3
**Presentation:** 3
**Contribution:** 2
**Rating:** 6
**Confidence:** 4

**Summary:**

This paper trains autoregressive Transformers on video datasets in a self-supervised manner, enabling the model to infer and imitate semantics from a demonstration video in a zero-shot way, allowing it to perform unseen tasks without fine-tuning.

**Strengths:**

1. This paper trained a transformer model called VidIT, which can learn motion information from a reference video and apply this motion to the prediction of a target video.

2. The method is effective, as extensive experiments demonstrate significant improvements in metrics, and the visual results further prove that the method effectively learns motion information.

3. This approach is based on an interesting discovery, which the authors define as "zero-shot video imitators." This finding could inspire future work in the transformer series.

4. The method can be applied to various tasks, such as unconditional video prediction, proving its strong generalization ability.

**Weaknesses:**

See questions

**Questions:**

1. However, the main contributions of this paper are not entirely clear to me. I believe there might be two key contributions:
a) The development of a video imitator model capable of mimicking actions from a reference video.
b) The demonstration that transformers possess zero-shot capabilities without needing specific training for tasks involving action imitation.

If the primary contribution is the first point, then the task of "imitating actions from a reference video" has been addressed in several previous works [1, 2]. Moreover, achieving the results shown in this paper could also be done using some existing open-source video editing models. Additionally, similar effects can be achieved by controlling camera angles [3].

If the primary contribution is the second point, then readers of this paper would be highly interested in understanding why transformers possess this zero-shot ability. However, the paper only demonstrates through experiments that task-specific training and the zero-shot approach yield similar performance. In my opinion, this is insufficient to explain the transformer's zero-shot capabilities. I recommend providing evidence through analyses of feature maps or attention scores to support this claim.

2. In Section 4.1, the paper introduces four settings but ultimately selects two. It is recommended that the names of these settings remain consistent throughout the paper to avoid confusing the readers. Also, ensure that these names are consistent with the names used in the experiments in Section 5.

3. Additionally, I suggest including more details about the training process, such as the hardware used and the approximate training time.

---

> ### Author Response · Authors · 2024-11-19
> **Responese to Reviewer PhVw**
>
> We sincerely thank Reviewer PhVw for the detailed and constructive comments. We would like to address the questions individually below:
>
> > Q1. The main contribution of the paper.
>
> We would like to clarify that the primary contribution of our work is to demonstrate that auto-regressive transformers can generalize to recover the semantics of demonstration videos without explicitly training on an imitation objective. We believe this ability, which we refer to as the **“zero-shot imitation ability”** in our paper, aligns more closely with the **second point** in your review.
>
> To verify the interpretability of this zero-shot imitation ability, we present an analysis of the **attention maps within the demonstration video in Appendix A.5 and Figure 11 in the paper**. This analysis focuses on the attention patterns when generating tokens. In Figure 11, the first row depicts the original frame sequence. The second row represents the training case, where frames are drawn from a single video clip. The third row illustrates the inference case, where the sequence begins from a different query frame.
>
> Through this comparison, we observe that the generated tokens in both cases exhibit consistent attention to previous frames. This consistency indicates that the model focuses on capturing the main semantic information (e.g., the movement of an object in this case) and can effectively reconstruct this information in a different scene to perform imitation.
>
> Moreover, we figures out how VidIT fails to generate expected contents by analyzing attention distribution which is shown in Figure 12 (Appendix A.5) inthe manuscript.  Successful generation samples show a clear tendency to allocate more attention to the demonstration frames, reflecting their ability to comprehend and leverage the semantic information provided. In contrast, failure cases exhibit attention curves with a stronger focus on the query frames, effectively disregarding the guidance from the demonstration.
>
>
>
> We believe this analysis reveals the intrinsic mechanism underlying the model's imitation capability and significantly strengthens our experimental findings.
>
> > Q2. The experimental settings.
>
> We would like to clarify that all the four settings including “No Demonstration”, “Random Demonstration”, “In-class Demonstration”, “Contrastive Demonstration” have been properly discussed in our original manuscripts. We compare the first three settings in Table 1 as our main experimental findings. And the comparison of contrastive demonstration is discussed in Table 2. We are sorry for the potential confusion to readers by the vague notation in the manuscript, and we have unified the notations in the revision.
>
> > Q3. Including more details about the training process, such as the hardware and the approximate training time.
>
> Thanks for the suggestion! We have added an additional subsection in the paper (appendix B.4: training details) to enhance reproducibility and provide necessary details for practitioners. To sum up, we trained VidIT on 2*8 NVIDIA H100 GPUs, and the training time for one epoch during pretraining is approximately 3 hours for the largest variant of the model (1.1B parameters). We utilized Distributed Data Parallel (DDP) integration via the PyTorch-Lightning trainer for efficient distributed training.
>
> We hope the response address your question. If you have any further inquiries, please feel free to reach out. We are open to continued discussion.

---

> > ### Comment · Reviewer_PhVw · 2024-11-20
> >
> > Thank you for your rebuttal, after reading this, I have some other doubts.
> >
> > I would like to discuss this with the author because you chose zero-shot capability as the paper's main contribution. When initially researching and designing the experiment, why did you choose autoregressive as the experimental model instead of other more popular generation frameworks in recent years (such as diffusion)?
> >
> > This is a good paper, I can give it 6 points at present. However, because the main contribution is to the study of zero-shot capability, as a reader I may be more concerned about whether this capability can be applied to other models. Therefore, I cannot give it a higher score.

---

> > > ### Author Response · Authors · 2024-11-22
> > > **Response to follow-up question**
> > >
> > > Thanks for your response and positive feedback! There are two main reasons why we choose an auto-regressive generative model.
> > >
> > > Firstly, instead of focusing solely on maximizing the quality of generative models, our primary motivation is to investigate whether video generative models can extract, understand, and utilize the semantics contained in context frames. With this goal in mind, the autoregressive Transformer model is a natural choice, given that textual LLMs have demonstrated zero-shot in-context learning capabilities.
> > >
> > > Secondly, from the empirical perspective, we evaluated the video imitation capacity of an open-source diffusion model Open-Sora Plan[1], by appending demonstration and query frames as the context and letting the model generate consequences. The results are presented in this anonymous site: https://anonymous.4open.science/w/VidImit-page-656D/#diffusion-results. We show that Open-Sora Plan tends to generate static contents from the last frame on the query. These results demonstrate that diffusion models struggle to comprehend semantics from demonstrations, primarily due to their lack of explicit autoregressive conditioning on previous frames. We will conduct a more thorough qualitative investigation in the revision. Exploring the underlying reasons for the varying performances of AR and diffusion models is an intriguing direction for future work.
> > >
> > > [1] Open-Sora-Plan. https://github.com/PKU-YuanGroup/Open-Sora-Plan/

---

### Official Review · Reviewer_G9JK · 2024-10-20

**Soundness:** 3
**Presentation:** 3
**Contribution:** 4
**Rating:** 8
**Confidence:** 4

**Summary:**

The paper introduces Video Imitator (VidIT), an autoregressive transformer model trained for zero-shot video imitation, enabling it to imitate the semantics of a given demonstration video and generate coherent video clips without requiring further fine-tuning. VidIT uses a next-token prediction objective to learn from video datasets, allowing it to generate video sequences that are semantically aligned with demonstrations. The model is evaluated on visual quality and semantic accuracy using both automatic metrics and human assessment.

**Strengths:**

1. Zero-shot Capability: VidIT achieves a zero-shot imitation capacity, enabling it to generalize from video demonstrations without further training.

2. Versatile Applications: The model can be applied to multiple downstream tasks, including robotic simulation, video planning, and segmentation.

3. Comprehensive Evaluation Benchmark: Both objective metrics and subjective human evaluations validate the quality and semantic coherence of generated videos.

4. Scalable Model: The model demonstrates improved performance when scaled up, providing strong semantic control in video generation.

**Weaknesses:**

1. Limited Sequence Length: Because now the mainstream video generation solutions have reached more than 5 seconds, the evaluation is primarily on short sequences; the performance on longer sequences remains untested.

2. Gap Between the Training Web Videos and Evaluation Metrics: In the benchmark constructed in this article, the domains of the evaluation metric are mostly in indoor scenes (SSv2), which is relatively limited. This is likely why adding more web videos to the training data did not make the model perform better. To get more accurate conclusions, evaluation metrics should be developed for more open scenarios and diverse tasks mentioned in the appendix, such as task imitator.

**Questions:**

How well does the model generalize to longer video sequences compared to the current evaluations on short clips?

---

> ### Author Response · Authors · 2024-11-19
> **Response to Reviewer G9JK**
>
> We appreciate Reviewer G9JK for highlighting our work’s contribution to the video generation research frontier.  We would like to address your questions below:
>
> > W1/Q1: Limited Sequence Length; How well does the model generalize to longer video sequences compared to the current evaluations on short clips?
>
> The generated sequence length is constrained by the context length of the backbone model (i.e., LLaMA). To address this, one approach is to utilize training-free techniques such as sliding window during inference to generate longer video clips. Please refer to the cases in MineRL in Figure 7 (also the videos in the demo page) for example. By fixing the demonstration frames and generating consequences with sliding window, our model can produce long video clips (e.g., 10 seconds) that follow the instructions in the demonstration video. Through our early experiments, we found that semantic accuracy of longer generated sequences remains on par with that of shorter ones. We concluded that the key factor influencing evaluation metrics is not the clip length, but rather the type of demonstrations used.
>
> In addition, the LLaMA backbone of VidIT makes it natural to utilize long-context extension techniques in LLMs such as YaRN [1] / LongRoPE [2]. Besides, considering the video modality nature, it is also natural to incorporate video tokenizers with both temporal and spatial down-sampling to create more compact video tokens, thereby enabling the generation of longer video sequences. We are currently working on both directions as part of the future work.
>
> [1] Peng, Bowen, et al. "YaRN: Efficient Context Window Extension of Large Language Models.", ICLR 2024.
>
> [2]  Ding, Yiran, et al. "LongRoPE: Extending LLM Context Window Beyond 2 Million Tokens.", ICML 2024.
>
> > Q2. Gap Between the Training Web Videos and Evaluation Metrics
>
> We appreciate your suggestion and agree that our conclusions could be further strengthened by evaluating on additional tasks. We would like to clarify that one of the criteria for selecting the evaluation datasets is the well-organized semantic label space, which allows us to construct semantically related or opposite samples for efficient testing. For example, action labels like “Push something from left to right” and “Push something from right to left” are clearly defined as semantically opposite. With this in mind, the final chosen evaluation set includes RT-1, SSv2, and Minecraft, which all have well-defined semantic spaces.
>
> Additionally, we are actively working on expanding the benchmarking of VidIT to include other applications, such as task success rate for RL agents and segmentation metrics for task imitators.
>
> We hope the response address your question. If you have any further inquiries, please feel free to reach out. We are open to continued discussion.

---

> ### Author Response · Authors · 2024-11-23
> **We hope that our response addresses your concern**
>
> Dear Reviewer G9JK,
>
> We greatly appreciate the time you've invested in reviewing our response. Having submitted our rebuttal, we are eager to know if our response has addressed your concern. As the end of the rebuttal phase is approaching (Nov. 26, 2024) , we look forward to hearing from you for any further clarification that you might require.
>
> Best,
>
> Submission 10648 Authors

---

> > ### Comment · Reviewer_G9JK · 2024-11-29
> > **Response by the reviewer**
> >
> > Thanks for your answer that cleared up my confusion. I will keep my score and you can focus on addressing others' opinions to increase the probability of raising the score.

---

### Official Review · Reviewer_vvMc · 2024-11-04

**Soundness:** 2
**Presentation:** 2
**Contribution:** 2
**Rating:** 6
**Confidence:** 4

**Summary:**

The authors proposed to train an autoregressive model for video imitation. The model training is performed in a self-supervised manner using a continuous video. Then, in the inference stage, given a demonstration and query video, the model could generate subsequent video frames of the query video that imitate the same semantics or actions conveyed by the demonstration video. The model shows zero-shot capacity when generating the imitation video frames.

**Strengths:**

- The topic of video imitation is valuable and would attract sufficient attention in the community, especially for embodied AI, since the approach could be applied to generate a large number of unseen training data, which is costly to obtain manually in the real world.
- It is interesting that the autoregressive model trained without using demonstration videos can generalise to the imitation inference stage when the demonstration is provided.
- The proposed framework is flexible and can be extended to imitate videos conditioned on text, which might be easier to acquire than demonstration videos and practically applicable.

**Weaknesses:**

- On the technical side, the novelty of the proposed method is limited. The combination of VQ-VAE and Transformer is commonly adopted for image generation.
- There is a lack of detailed explanation about why self-supervised training without acquiring demonstrations during training will generalise to video imitations given demonstrations.
- The visual quality of the generated imitation video is limited as shown in Table 1. The V-Acc and P-Acc scores are quite low, suggesting that the generated video's content might differ from the expected one. So, the practical usage of the proposed method is questionable.
Failure cases would help readers understand the method's limitations. I suggest the authors provide a detailed analysis accordingly.
- The evaluation of using text as conditions is very limited.
- The quality of writing could be further improved by providing more details regarding task definition and technical details. For example, it would be better to list all the types of demonstration videos. Currently, it is unclear if any more tasks are involved other than object manipulation and camera movement (and their detailed types). The training objectives could be clarified (currently missing) for those unfamiliar with VQ-VAE/VQ-GAN for better reproducibility.

**Questions:**

- It is confusing to define a contrast action. What is the discipline? It might be better to provide a detailed list of all the situations that are considered.
- Is it possible to provide the V-acc and LPIPS scores in Table 2? In my view, V-acc and LPIPS are more faithful and direct measurements of the visual content of the generated videos.

---

> ### Author Response · Authors · 2024-11-19
> **Response to Reviewer vvMc (1/2)**
>
> We appreciate Reviewer vvMc for the detailed and constructive comments. We would like to address the issues individually below:
>
> > W1. The novelty of the proposed method is limited. The combination of VQ-VAE and Transformer is commonly adopted for image generation.
>
> We agree that VQ-VAE + Transformer is a common pipeline used in image generation, however, we would like to clarify that illustrating such paradigm as a zero-shot video imitator is not a trivial practice. We choose the auto-regressive generation as it is smoothly aligned with our task where a prompting video clip is given, different from generating a complete video from scratch. We also have made some specific designs on the architecture such as data compositions. In addition, we show that this decoder-only architecture allows for the scalability on both of the visual quality and semantic accuracy. As a conclusion, we did not straightforwardly follow the common practice in image generation, but design the pipeline to verify our hypothesis.
>
> > W2. Lack of detailed explanation about why self-supervised training can generalize to video imitation.
>
> We present an analysis of the **attention maps within the demonstration video in Appendix A.5 and Figure 11 in the paper**. This analysis focuses on the attention patterns when generating tokens. In Figure 11, the first row depicts the original frame sequence. The second row represents the training case, where frames are drawn from a single video clip. The third row illustrates the inference case, where the sequence begins from a different query frame.
>
> Through this comparison, we observe that the generated tokens in both cases exhibit consistent attention to previous frames. This consistency indicates that the model focuses on capturing the main semantic information (e.g., the movement of an object in this case) and can effectively reconstruct this information in a different scene to perform imitation. Moreover, we dive into how VidIT fails to generate expected contents by analyzing attention distribution which is shown in Figure 12 (Appendix A.5) in the manuscript.  This analysis also explores the inner mechanism of our model, which shows the different attention allocation for demonstration frames and query frames. The detailed discussion can be found in response to W3 below.
>
> We believe this analysis reveals the intrinsic mechanism underlying the model's imitation capability and significantly strengthens our experimental findings.
>
> > W3. Visual quality is limited. V-Acc and P-Acc scores are quite low. Failure cases would help readers understand the method's limitations.
>
> We would like to clarify that the V-Acc of around 25% is restricted by the pretrained video classifier we adopt (`videomae-base-short-finetuned-ssv2`). It is challenging to achieve a high classification accuracy on the sophisticated SSv2 dataset with 174 classes, and the accuracy in 8 frames on the ground truth SSv2 valid set is about 40~%, which can be seen as the upperbound for V-Acc, as the generation results with even tiny blurs will hurt the classification accuracy. P-Acc is also a validation of a classifier with 174 classes, so it mirrors the problem in V-Acc. In addition, instead of the absolute accuracy number, we validate our idea mainly by the relative comparison of the metrics among different demonstration settings, where the in-class demonstration outperforms the other two settings.
>
> For the failure case issue, we think this is a great suggestion and thanks for that! We have analyzed a set of failure cases and concluded that the model shows limited control when given semantically unrelated demonstrations. As shown in Appendix A.4 and Figure 10 in the paper, when prompted with semantically unrelated videos, the model fails to generate consequences of the query that follows the semantics in demonstrations.
>
> For instance, in the first row, the demonstration video conveys the semantic information of picking up a ball and moving it downward. However, when starting with a query video depicting an unrelated action, such as folding a blanket, the model struggles to reconcile the two contexts.
>
> Moreover, we extend the analysis of failure cases by examining the differences in attention distribution, as shown in Figure 12 (Appendix A.5) of the manuscript.  Successful generation samples show a clear tendency to allocate more attention to the demonstration frames, reflecting their ability to comprehend and leverage the semantic information provided. In contrast, failure cases exhibit attention curves with a stronger focus on the query frames, effectively disregarding the guidance from the demonstration.

---

> > ### Author Response · Authors · 2024-11-19
> > **Response to Reviewer vvMc (2/2)**
> >
> > Follows up to Response to Reviewer vvMc (1/2).
> >
> > > W4. Evaluation of using text as conditions is very limited.
> >
> > The primary motivation of our work is to explore a vision generative model’s ability to recover semantics conditioned on video demonstrations. Accordingly, the main narratives and experiments are designed with this purpose in mind. While text conditioning is not the central focus of our paper, we included it to demonstrate the multi-modal capabilities of the video generation model. From the results, we successfully show that the semantics within the text can effectively guide the generation of video content. We fully agree that exploring how textual information interacts with video demonstrations presents a promising research direction. Investigating this interplay could further enhance our understanding of multi-modal conditioning and improve video generation performance in more complex scenarios.
> >
> > > W5. The quality of writing could be improved by providing more details;  Currently, it is unclear if any more tasks are involved;  The training objectives could be clarified for VQ-VAE.
> >
> > We appreciate your suggestions for making the paper easier and clearer with readers. For the list of demonstrations, we provide all the types of demonstration in this anonymous link: https://anonymous.4open.science/r/VidImit-page-656D/rebuttal/all_classes.json. It has 174 classes in total which is the labels for Something-something v2 dataset. Besides the mentioned classes, we also show generated cases on other types of demonstrations in our anonymous website and appendix which also perform well. Moreover, the experiments on other test sets such as robotic data and Minecraft also shows our model's efficacy.
> >
> > For the technical details, we have added a section in Appendix B.4, which includes information on the GPUs used, training time, and parallel training configuration.  For the clarification on VQ-VAE/VQ-GAN, we provide a clear depiction of our model architecture, including VQ tokenizer components, in Figure 2, along with a detailed interpretation in lines 206–208. We adopted a public pre-trained VQ-VAE and kept it fixed during training, as specified in lines 203–204 of the paper. The training objective corresponds to the cross-entropy loss for next-token prediction in the downstream LLaMA model, as described by the negative log loss for each factor in Equation (3).
> >
> > > Q1. How to define a contrast action?
> >
> > We have mentioned that there are 174 classes in total in the Something-something v2 benchmark. Among these classes, we manually choose 20 classes that are entirely opposite in semantics. The contrastive pairs are like ‘Moving [something] down’ and ‘Moving [something] up’, ‘Pulling [something] from left to right’ and ‘Pulling [something] from right to left’. We also provide all the contrastive classes in the anonymous link: https://anonymous.4open.science/r/VidImit-page-656D/rebuttal/contrastive_classes.json .
> >
> > > Q2. Is it possible to provide the V-acc and LPIPS scores in Table 2?
> >
> > Yes, the V-Acc and LPIPS scores were evaluated alongside the metrics presented in Table 2. For the formatting constraint, we only show P-Acc as the representative semantic accuracy metric and PSNR as the representative visual quality metric in the current manuscript. The V-Acc and LPIPS scores of these models are included in the table for completeness.
> >
> > | Baselines | V-Acc | P-Acc | PSNR  | LPIPS | FID   | FVD    |
> > | --------- | ----- | ----- | ----- | ----- | ----- | ------ |
> > | LVM       | 21.2  | 27.1  | 12.45 | 0.489 | 27.58 | 163.13 |
> > | DeLVM     | 24.1  | 32.8  | 12.69 | 0.462 | 23.65 | 132.09 |
> > | LWM       | 21.8  | 24.9  | 11.89 | 0.474 | 29.91 | 191.56 |
> > | **VidIT (ours)**    | **25.9**  | **38.5**  | **13.07** | **0.450** | **22.30** | **126.32** |
> >
> > From the results, it is evident that the VidIT model consistently outperforms the baseline models in video imitation tasks, reinforcing its superior capability in capturing both semantic alignment and visual quality.
> >
> > We hope the response address your question. If you have any further inquiries, please feel free to reach out. We are open to continued discussion.

---

> > ### Author Response · Authors · 2024-12-02
> >
> > Dear Reviewer vvMc,
> >
> > As the discussion period is approaching the last day, please let us know if we have addressed your concerns. And we are welcome for further discussions. We greatly appreciate the time you've invested in reviewing our paper!
> >
> > Best,
> >
> > Submission 10648 Authors

---

> > > ### Comment · Reviewer_vvMc · 2024-12-02
> > > **most of my concerns have been addressed**
> > >
> > > I want to thank the authors for clarifying my questions, and I think most of my concerns have been addressed. Hence, I will increase my rating to 6. The only problem still is the practical usage in real-world cases, given the relatively low quality (e.g., the demonstrated blurry) of the generated video. It would be great if the authors could discuss its applicability further.

---

> ### Author Response · Authors · 2024-11-23
> **We hope that our response addresses your concern**
>
> Dear Reviewer vvMc,
>
> We greatly appreciate the time you've invested in reviewing our response. Having submitted our rebuttal, we are eager to know if our response has addressed your concern. As the end of the rebuttal phase is approaching **(Nov. 26, 2024)** , we look forward to hearing from you for any further clarification that you might require.
>
> Best,
>
> Submission 10648 Authors

---

> ### Author Response · Authors · 2024-11-29
>
> Dear Reviewer vvMc,
>
> We greatly appreciate your time and effort in reviewing our work and sincerely apologize for the delayed response. We are eager to ensure that we have adequately addressed your concerns and are prepared to offer further clarifications or address any additional questions you may have. As the end of discussion is approaching (Dec.3, 2024), we would be grateful if you could share your thoughts on our rebuttal.
>
> Best,
> Submission 10648 Authors

---

> ### Author Response · Authors · 2024-12-03
> **Follow-up response**
>
> Dear Reviewer vvMc,
>
>
> We are happy that our responses have addressed most of your concerns! Regarding real-world applications, we have illustrated the performance of VidIT model on robotic datasets as presented in our paper, which demonstrates the potential for our model to be extended to a variety of real-world scenes. To further enhance the model's capabilities, we are actively exploring the adoption of more advanced video tokenizers and larger transformer architecture as a part of future work.
> In addition to functioning as an explicit video generator, our model can also serve as the core engine within a real-world simulator, as described in Appendix A.1. In this setup, the generated video is converted into action signals via the inverse dynamic mechanism, which are directly executed by the environment to provide consequences. This setting utilizes the semantic information conveyed in generated videos while mitigating the impact of the generation quality.
>
> We hope the responses address your questions. Thank you for your valuable feedback and thoughtful suggestions!

---

### Official Review · Reviewer_vT7n · 2024-11-04

**Soundness:** 3
**Presentation:** 3
**Contribution:** 3
**Rating:** 8
**Confidence:** 4

**Summary:**

This paper proposes a video generation based on autoregressive transformers trained with the objective of imitating a set of seed videos. In experiments, the authors show that the model is able to imitate tasks presented to it in an in-context manner. In addition, the learned representations coming from the model are also useful for classification tasks.

**Strengths:**

Originality:
The idea of performing in-context autoregressive video generation is interesting and I have not seen it being done before. Autoregressive video generation without text is very difficult, and this approach seems to address some of the difficulties.

Quality / Clarity:
The paper is clear and easy to read. The proposed idea is simple and I believe easy to replicate from the descriptions.

Significance:
I believe that even though this paper is not producing mind blowing results such as Sora, Kling, Veo and others. The ideas proposed here can be useful for the scientific community to continue exploring the problem of video generation

**Weaknesses:**

Weaknesses / Questions:

Missing baselines:
* Not sure if I missed this from reading the Baselines paragraph in the manuscript, but did the authors include a simple next frame prediction baseline?

**Questions:**

See weaknesses.

---

> ### Author Response · Authors · 2024-11-19
> **Response to Reviewer vT7n**
>
> We sincerely thank Reviewer vT7n for considering our work interesting and brings new perspective to the video generation community. We would like to address the question below:
>
> > Q1. Did the authors include a simple next frame prediction baseline?
>
> We have included a next-frame prediction baseline in the original manuscript. This baseline is referred to as "No Demonstration" in lines 282-283 of the manuscript, and the corresponding results can be found in Table 1 on the rows starting with “No”. This setting is equivalent to a simple next-frame prediction, where the model is expected to complete the query without any demonstration.
>
> By comparing with this baseline, we show that using semantically unrelated videos in the demonstration leads to a decrease in semantic accuracy (e.g., “No Demonstration” vs. “Random Demonstration”). In contrast, the performance improves significantly when the generation process is guided by semantically closer demonstrations (e.g., “No Demonstration” vs. “In-class Demonstration”). We are sorry for the potential confusion to readers by the vague notation in the manuscript, and we have unified the notations in the revision.
>
> We hope the response address your question. If you have any further inquiries, please feel free to reach out. We are open to continued discussion.

---

> ### Author Response · Authors · 2024-11-23
> **We hope that our response addresses your concern**
>
> Dear Reviewer vT7n,
>
> We greatly appreciate the time you've invested in reviewing our response. Having submitted our rebuttal, we are eager to know if our response has addressed your concern. As the end of the rebuttal phase is approaching **(Nov. 26, 2024)** , we look forward to hearing from you for any further clarification that you might require.
>
> Best,
>
> Submission 10648 Authors

---

> > ### Comment · Reviewer_vT7n · 2024-11-26
> > **Next frame prediction baseline**
> >
> > Thanks for your clarification. I have increased my score. Nice job!

---

### Author Response · Authors · 2024-11-19
**Global Response**

We sincerely thank all the reviewers for their time and effort in reviewing our work. We are encouraged to see that our work is acknowledged to "be interesting"(Reviewer vT7n, vvMc, PhVw), "be useful to the community" (Reviewer vT7n, vvMc), "widely applicable" (Reviewer G9JK, vvMc, PhVw) and "have extensive experiments" (Reviewer G9JK, PhVw).

Taking into account all the reviewers' comments, we have provided individual responses to each reviewer. Furthermore, we have added additional analysis and experiments in the paper's appendix. The main body of the paper remains unchanged, except for clarifying the notations in the first column of Table 1. Below is a brief summary of the added contents:

* **Appendix A.4**: Failure cases. This section presents failure cases where VidIT fails to align the query video with the semantic context of the demonstration video. **Mentioned: Reviewer vvMc's Weakness 4.**
* **Appendix A.5:** Interpretability of Generalization. This section highlights the interpretability of VidIT and its ability to generalize to imitation tasks in a zero-shot manner. Also companied with a failure cases analysis on attention distribution. **Mentioned: Reviewer vvMc's Weakness 4, Reviewer PhVw's Question 1.**
* **Appendix A.6:** Full Evaluation Metrics on Baselines. This section completes Table 2's results with the all the evaluation metrics. **Mentioned: Reviewer vvMc's Question 2.**
* **Appendix B.4:** More training details for training VidIT. **Mentioned: Reviewer vvMc's Weakness 5, Reviewer PhVw's Question 3.**

We hope these updates and clarifications address the reviewers' concerns and further strengthen our paper's quality.

---

### Meta-Review · Area_Chair_opJX · 2024-12-19

**Metareview:**

The paper demonstrates the zero-shot video imitation capabilities by training self-supervised autoregressive Transformers on video data. Such in-context capabilities enable solving unseen tasks by watching video demonstrations.

The rebuttal is well-accepted by most reviewers. After rebuttal, all reviewers are positive about this paper and two are enthusiastic.
As raised by reviewers, a deeper investigation and analysis on varied models/architectures and real-world usage will strengthen this paper further.

The area chair recommends accepting this paper. The zero-shot video imitation capabilities provide insights into research on exploring GPT-3 like in-context learning capabilities in video generation and embodied tasks.

**Additional Comments On Reviewer Discussion:**

### Main concerns raised before rebuttal
- Reviewer vT7n (score 6): missing next-frame prediction baseline
- Reviewer vvMc (score 5): the novelty of the architecture, limited explanation about why the model can generalize to video imitation, poor visual generation quality.
- Reviewer G9JK (score 8): limited sequence length and time duration, gap between training and evaluation.
- Reviewer PhVw (score 5): interpretability of this zero-shot imitation ability, more details about the training and environment, capabilities cross models.

### After rebuttal
- Reviewer vT7n increases score from 6 to 8.
- Reviewer vvMc increases score from 5 to 6.
- Reviewer G9JK maintains a score of 8.
- Reviewer increases the score from 5 to 6.

The remaining potential improvements include a deeper investigation and analysis on varied models/architectures and real-world applications. The authors have made responses by adding more discussion.

Overall,  I think most of the concerns are addressed. All reviewers are positive about this paper.

---

### Decision · Program_Chairs · 2025-01-22

Accept (Poster)